

# Influence of Atmospheric Circulation on the Interannual Variability of Transport from Global and Regional Emissions into the Arctic

Cheng Zheng[1], Yutian Wu[1], Mingfang Ting[1,2], Clara Orbe[3]

[1]Lamont-Doherty Earth Observatory, Columbia University, Palisades, NY, 10964, USA
[2]Columbia Climate School, Columbia University, New York, NY
[3]NASA Goddard Institute for Space Studies, New York, NY, 10025, USA

*Correspondence to*: Cheng Zheng (czheng@ldeo.columbia.edu)

**Abstract.** Trace gases and aerosols play a crucial role in shaping Arctic climate through their impacts on radiation and chemistry. The concentration of these substances over the Arctic is largely determined by long-range transport originating
from midlatitude and tropical source regions. In this study, we explore how atmospheric circulation modulates the interannual variability of long-range transport into the Arctic by utilizing a chemistry climate model. Idealized tracers, which have fixed lifetimes and spatially varying but temporally fixed surface emission corresponding to the climatology of anthropogenic emissions of the year 2000, are employed to isolate the role of atmospheric transport from emission and chemistry in modulating interannual variability. Tracers emitted from different source regions are tagged to quantify their
relative contributions. Model simulations reveal that tracers from Europe, East Asia, and North America contribute the most to Arctic tracer mass, followed by those from the Tibetan Plateau and South Asia, and the Middle East. These regional tracers are predominantly transported into the Arctic mid-to-upper troposphere, with the exception of tracers from Europe during winter, which are transported into the Arctic lower troposphere. Our analysis shows that the interannual variability of transport into the Arctic for each regional tracer is determined by the atmospheric circulation over the corresponding
emission region, i.e., anomalous poleward and eastward winds over the source region promote transport into the Arctic. Considering tracers with global emissions, a southward shift of the midlatitude jet during winter favours increased transport into the Arctic, particularly for tracers emitted over Asia, aligning with previous studies. Comparisons of tracers with different lifetimes indicate that the interannual variability of shorter lifetime tracers is predominantly influenced by regional tracers with shorter transport pathways into the Arctic (e.g., Europe), while the interannual variability of longer lifetime
tracers is more contributed by regional tracers with higher emissions (e.g., East Asia).





# 1 Introduction

In recent decades, rapid warming has been observed in the Arctic, with a warming rate significantly faster than the global average (e.g., IPCC, 2021). Trace gases and aerosols, primarily transported into the Arctic from emission sources in the midlatitude and the tropics (e.g., Bottenheim et al., 2004; Fisher et al., 2010; Klonecki et al., 2003; Kupiszewski et al., 2013; Law and Stohl, 2007; Rahn and McCaffrey, 1980; Shindell, 2007; Shindell et al., 2008; Stohl, 2006), have substantial impacts on Arctic chemistry, as well as Arctic climate via their radiative influences. Shortwave radiative transfer can be modulated by black carbon in the Arctic, as it absorbs sunlight (e.g., Quinn et al., 2006; Warren and Wiscombe, 1980) when deposited onto the Arctic surface or when suspended in the Arctic haze layer, which has been suggested to execute twice as large a climate forcing as carbon dioxide over the Arctic (Hansen and Nazarenko, 2004). Aerosols also modulate longwave radiative transfer, particularly via their indirect effects as they affect on the microphysical properties of clouds (Coopman et al., 2018; Garrett and Zhao, 2006; Lubin and Vogelmann, 2006). Additionally, halocarbons originating from midlatitudes influence the production of tropospheric ozone in the Arctic (Atlas et al., 2003; Klonecki et al., 2003), which acts as a greenhouse gas. These influences highlight the importance of comprehending the transport processes responsible for bringing trace gases and aerosols from the midlatitudes and the tropics into the Arctic.

The observed concentration of trace gases and aerosols in the real atmosphere is influenced by emission, transport and removal processes. Modelling studies are usually applied to disentangle their respective roles. Specifically, to isolate the role of long-range atmospheric transport, tracers with idealized emission and chemical removal processes have been implemented into climate models. Recently, such tracers have been utilized in the Chemistry Climate Model Initiative (CCMI; Eyring et al., 2013; Morgenstern et al., 2017) to identify and compare the characteristics of atmospheric long-range transport among chemistry-climate models. Orbe et al., (2018) conducted an analysis of tracers with zonally uniform emission and prescribed lifetimes in the CCMI models, attributing inter-model differences in transport into the Arctic to variations in model's midlatitude convective transport, particularly over the ocean in winter. However, zonally uniform emission does not represent real world emissions. Thus, an idealized "CO50" tracer implemented in multiple CCMI models, with spatially varying but temporally constant surface emission reflecting realistic anthropogenic carbon monoxide (CO) emissions and a fixed decay time of 50 days, better represents the transport characteristics with more realistic emissions. Yang et al., (2019) attributed the inter-model spread of CO50 transport into the Arctic to differences in the meridional location of the model simulated Hadley Cell edge (or the jet stream).

The long-range transport using idealized tracers from different parts of the globe into the Arctic has also been investigated in previous studies. Orbe et al., (2015) identified the origin of Arctic airmass by tagging the airmass with the region where it last contacts the planetary boundary layer. They found that the Arctic airmass in the lower, middle, and upper troposphere is mostly contributed by airmass originated from the Arctic, midlatitudes, and tropics respectively. The contribution from midlatitude regions primarily comes from the oceans during winter, while the contribution during summer largely comes from Asia and North America. Zheng et al., (2021) investigated the transport pathways into the Arctic and associated



timescales from different regions in the northern hemisphere. They found midlatitude emitted pulse tracers can be transported into the Arctic via transient eddies in about a week; the tracers can also be picked up by the midlatitude jet and then transported into the Arctic over the jet exit regions in the North Pacific and North Atlantic. Both Orbe et al., (2015) and Zheng et al., (2021) utilized spatially uniform "emissions", which does not accurately represent realistic emissions that often have high spatial variations. We will identify the role of different regions by utilizing more realistic emissions.

Interannual variability in atmospheric circulation also leads to variations in tracer transport into the Arctic. Previous studies have highlighted the role of biomass burning emission on Arctic CO interannual variability (e.g., Monks et al., 2012), with the variability in emission significantly modulated by the El Niño–Southern Oscillation (ENSO). In examining the impact of atmospheric circulation variability, the North Atlantic Oscillation (NAO) has been identified as a significant modulator of transport into the Arctic (Eckhardt et al., 2003; Duncan and Bay 2004; Octaviani et al., 2015). Specifically, the positive

phase of NAO enhances the transport of pollution emitted in Europe into the Arctic. Despite earlier efforts, a systematic understanding of how atmospheric circulation modulates the interannual variability of transport into the Arctic is lacking. In this study, we employ tracers similar to CO50 discussed above, with temporally fixed but spatially realistic anthropogenic emission, to understand the role of atmospheric circulation in tracer transport into the Arctic, particularly regarding interannual variability focusing on the following questions: 1) How does the atmospheric circulation modulate the transport

of tracers emitted by different source regions? 2) How does the relative contribution of each source region to the Arctic vary from year to year, and what is the associated circulation variability? 3) How does tracer lifetime impact the interannual variability of the transport?

The structure of the paper is summarized as follows: Sect. 2.1-2.2 provides an overview of the chemistry-climate model and the tracers employed, as well as how tracers are tagged by different emission regions. Sect. 2.3 describes the method for

analyzing the interannual variability of tracer transport into the Arctic. Basic characteristics of tracer transport into the Arctic from different emission regions in different seasons are presented in Sect. 3. In Sect. 4, the interannual variability of regionally and globally emitted tracers during winter and summer is explored. The influence of tracer lifetime is discussed in Sect. 5. The conclusions are presented in Sect. 6.

## 2 Methods

### 2.1 Model Simulations

The Whole Atmosphere Community Climate Model Version 6 (WACCM6; Gettelman et al., 2019), the high-top atmospheric component of the Community Earth System Model Version 2 (CESM2; Danabasoglu et al., 2020), is used to simulate the transport of idealized tracers. WACCM6 has 70 vertical levels with a horizontal resolution of 0.9° latitude and 1.25° longitude. We use fixed external forcing (i.e. greenhouse gases, anthropogenic aerosols) from the year 2000 and

integrate the model with prescribed year-to-year varying sea surface temperature (SST) and sea ice concentration (SIC) from 1960 to 2008 (a 49-year simulation). The SST and SIC boundary forcing is a merged product based on the Hadley Centre sea





ice and SST dataset version 1 (HadISST1) and version 2 of the National Oceanic and Atmospheric Administration (NOAA) weekly optimum interpolation SST analysis (Hurrell et al., 2008) designed for CESM atmospheric-only simulations. A 5-year simulation with SST and SIC of 1960 is performed first to spin up the model.

**2.2 Tracers**

We implement the idealized CO50 tracer into our simulation to quantify large-scale transport from different source regions to the Arctic, which has been used in multiple models participating in the Chemistry Climate Model Initiative (CCMI; Eyring et al., 2013; Morgenstern et al., 2017) to characterize atmospheric long-range transport. The emission of CO is a time-invariant flux at the surface (Fig. 1), corresponding to the annual mean value of anthropogenic emissions of carbon

monoxide (CO) for the year 2000, obtained from the Hemispheric Transport of Air Pollution (HTAP) REanalysis of the TROpospheric chemical composition (RETRO) (Eyring et al., 2013). The decay rate of CO50 in the atmosphere follows a spatially uniform 50-day e-folding decay time. The emission and loss rate of CO50 are both fixed in time, meaning that the variations in the spatial distribution of CO50 in the simulations are only driven by variations in atmospheric circulation. Note that CO50 only includes anthropogenic emission but not biomass burning, which is an important source of surface emission

for CO in the real atmosphere with strong spatial and temporal variations.

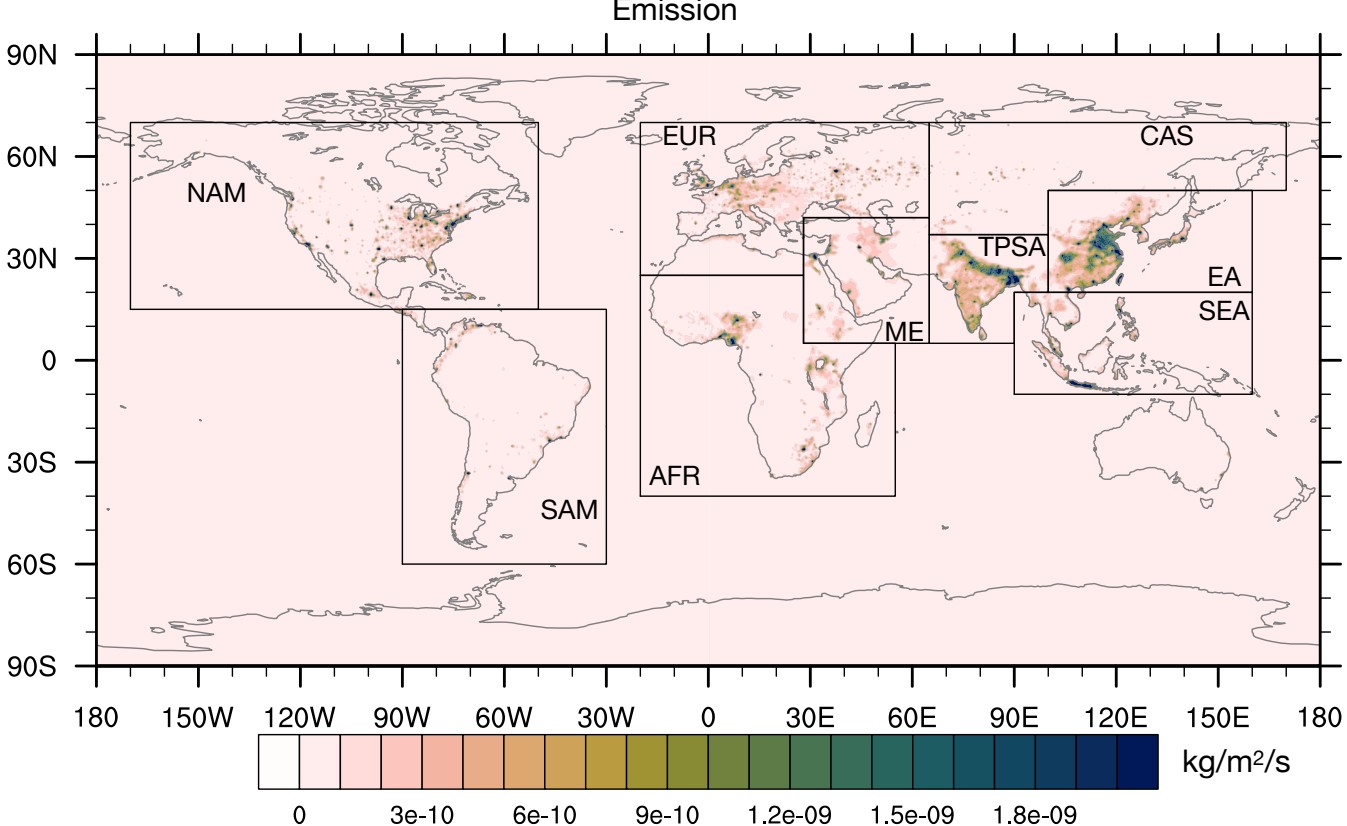



**Figure 1: Surface emission of CO50 tracers. The boxes show the boundaries of different regions for tagged tracers.**

To characterize the contribution to tracer concentration in the Arctic region from different emission regions, we tagged
tracers emitted from the following regions shown in Fig. 1: East Asia (EA), Southeast Asia (SEA), Central Asia and Siberia
(CAS), Tibetan Plateau and South Asia (TPSA), Europe (EUR), Middle East (ME), Africa (AFR), North America (NAM),
and South America (SAM). Any emissions outside the regions mentioned above are tagged as the remaining region (REM).
The tracers emitted from all regions (without regional tagging) will be referred to as global (GLB) tracers. Note that the sum
of contributions of all tagged regional tracers in the atmosphere has very little difference from the GLB tracer (not shown),
allowing for the attribution of individual regional contributions to the overall GLB tracer.

To explore whether different tracer lifetimes modulate the contribution from different regions to tracer concentration in the
Arctic, we also implement tracers similar to CO50 introduced above with the same surface emission flux but different
idealized lifetimes, including: CO100, CO25, CO15, CO10, and CO05, corresponding to lifetimes of 100, 25, 15, 10, and 5
days. Idealized tracers with different lifetimes could reflect the transport characteristics of different species of tracers in the
real atmosphere, such as butane, propane, and CO, which have lifetimes of 1 week, 2 weeks, and 2 months in the
troposphere, respectively. We will focus on the transport characteristics and interannual variability of CO50 in Sect. 3 and 4,
as it corresponds well with the realistic anthropogenic emission and the lifetime of CO. Tracers with other lifetimes will only
be discussed in Sect. 5.

## 2.3 Analysis of Tracer Transport into the Arctic

We explore tracer transport and the associated interannual variability into the Arctic during four seasons: winter (December
to February; DJF), spring (March to May; MAM), summer (June to August; JJA), and autumn (September to November;
SON), with the main focus on the winter and summer seasons. The Arctic region is defined as north of 70°N. We use Arctic
tracer mass as a metric to quantify tracer transport, which is the tracer mass integrated over 70°-90°N from the surface to 10
hPa. When investigating the horizontal structure of tracer transport, we consider the column integrated tracer mass (from
surface to 10 hPa) at each grid point. But when examining the vertical structure of tracer transport, we interpolate the tracer
data from the model hybrid levels to 28 pressure levels from 1,000 to 50 hPa. Interannual variability of the horizontal
structure of the column tracer mass in the Arctic is examined by Empirical Orthogonal Function (EOF) analysis. The EOF
analysis is conducted on the seasonal mean time series of column tracer mass over grid boxes within the Arctic region (north
of 70°N), and the covariance matrix is weighted by the area of each grid box during the analysis.





## 3 Characteristics of Tracer Transport from Different Emission Regions into the Arctic

### 3.1 Arctic Tracer Mass

The climatology of Arctic tracer mass across various seasons and emission regions is summarized in Fig. 2a-d. Among different emission regions, EA, EUR, and NAM stand out as the top three regions with the largest contribution. The three regions have similar contributions across different seasons, with the EUR tracer making the largest contribution during winter and the EA tracer contributing the most during other seasons. TPSA and ME are the next two regions with substantial contributions to the Arctic, with the rest of the 5 regions making relatively small contributions.

**Figure 2:** Summary of seasonal-mean Arctic tracer mass climatology of different regions (a-d), Arctic tracer mass climatology normalized by emission from each region (e-h), and interannual standard deviation of seasonal-mean Arctic tracer mass time series (i-l), during winter (DJF), spring (MAM), summer (JJA, and autumn (SON), respectively.



To determine the transport efficiency into the Arctic, i.e., the amount of tracer transported into the Arctic per unit emission, we normalize the total amount of the Arctic tracer mass from each region by the region's emission rate (regional emission per second), as displayed in Fig. 2e-h. Consistently across different seasons, EUR and CAS show the highest transport

efficiency, while NAM, EA, ME, and TPSA rank from third to sixth in terms of transport efficiency, respectively. The high transport efficiency of the EUR and CAS tracers is likely due to their closer proximity to the Arctic, meaning that less poleward transport is needed to convey the tracers into the Arctic, compared to other regions. This also means shorter transport time scales, resulting in less decay of the tracer along the path and higher transport efficiency. This will be further discussed in Sect. 3.2.1.

The interannual variability of the Arctic tracer mass for different emission regions, quantified by the interannual standard deviation of the seasonal-mean Arctic tracer mass, is shown in Fig. 2i-l. Not surprisingly, the emission regions with a larger climatology also have greater interannual variability, as EA, EUR, and NAM emerge as the top three regions with the largest interannual variability, followed by TPSA and ME. Interestingly, TPSA has almost the same magnitude of interannual variability as NAM during winter, though the actual tracer mass climatology for TPSA is less than half of that for NAM.

Overall the magnitude of the interannual variability (standard deviation) is on the order of 5-15% of the climatology across different regions and seasons. In the following analysis, we will place emphasis on the winter and summer seasons and focus on emission regions with large tracer contributions to the Arctic.

## 3.2 Spatial Structure of Transport

### 3.2.1 Horizontal Structure

To illustrate the horizontal structure of tracer transport, we present the climatological column tracer mass for tracers emitted from different regions during winter and summer in Fig. 3. 500 hPa climatological wind during winter and summer is illustrated by the vectors. For the GLB tracer during winter (Fig. 3a), the largest column tracer mass is concentrated over regions with high emissions, such as East Asia and South Asia. Transport of the tracers is characterized by eastward transport downstream from the jet stream in the extratropics. The EUR tracer (Fig. 3c) can spread into the Arctic over

Eurasia likely by transient eddies (Zheng et al., 2021) following this eastward transport. The EA, NAM and TPSA tracers (Fig. 3b, d-e) are first picked up by the North Pacific or North Atlantic jet streams, before being transported poleward into the Arctic, consistent with Zheng et al., (2021). Horizontal structures of the transport from the emission regions with smaller contributions to the Arctic (SEA, AFR, SAM, and REM) are shown in Fig. S1.

As discussed in Sect. 3.1, the high transport efficiency into the Arctic of EUR and CAS tracers is attributed to their closer

proximity to the Arctic. Notably, the transport distance into the Arctic is shorter for NAM compared to EA and TPSA. In addition, the NAM tracer is mostly concentrated in the extratropical and polar regions, while EA and TPSA tracers exhibit substantial equatorward transport into the tropics (Fig. 3). These factors contribute to the higher transport efficiency of the NAM tracer compared to the EA and TPSA tracers (Fig. 2e-h).





**Figure 3: Climatological column tracer mass (kg m$^{-2}$) during winter (a-g) and summer (h-n) for GLB, EA, EUR, NAM, TPSA, ME and CAS tracers. The white arrows depict climatological wind at 500 hPa. The scaling for the arrows is shown at the bottom right of the panel g) and n), unit in m s$^{-1}$. The purple circle depicts the boundary of the Arctic at 70°N.**

The primary features of the transport structures during summer closely resemble those in winter. Nevertheless, there are some notable differences, including: 1) More equatorward transport of the EUR tracer during summer compared to winter, leading to a lesser amount of EUR tracer in the Arctic during summer, which is also evident in Fig. 2; 2) there is less equatorward transport into the tropics for the EA and TPSA tracers during summer compared to winter, which only results in higher tracer concentrations in the subtropics and midlatitudes during summer, but no significant differences in the Arctic between winter and summer. The seasonality of the EUR tracer transport into the Arctic can be explained by the seasonality of the climatological wind. During summer, the equatorward wind over the Mediterranean region, associated with the divergence at the exit of the North Atlantic jet, promotes equatorward transport of the EUR tracer (Fig. 3j) into the subtropics, resulting in less transport of the EUR tracer into the Arctic. This equatorward transport into the subtropics is suppressed during winter (Fig. 3c) as the jet is stronger and shifts equatorward, meaning a higher amount of the tracer is transported downstream by the jet, which could then be transported into the Arctic.

### 3.2.2 Vertical Structure

To illustrate the vertical structure of the transport, the zonal mean tracer mixing ratio is shown as a function of vertical levels and latitudes in Fig. 4. The highest mixing ratio of the GLB tracer in the Arctic shifts from the mid-to-lower troposphere during winter (Fig. 4a) to the upper troposphere during summer (Fig. 4f). During winter, the EUR tracer (Fig. 4c) is directly transported into the Arctic in the lower troposphere, whereas the EA, NAM, and TPSA (Fig. 4b, d-e) tracers are first transported vertically upwards into the mid-troposphere and then horizontally into the Arctic. These features align with the findings in Orbe et al., (2015; see their Fig. 5), as midlatitude emitted tracers are primarily transported along the isentropes into the Arctic above the "polar dome" (Klonecki et al. 2003; Law and Stohl 2007) during winter. The poleward transport of the EUR tracer in the lower troposphere resembles the features of airmass origin from the northern high latitudes as in Orbe et al., (2015), which has a dominant contribution in the Arctic lower troposphere. This is reasonable given many of the emission hotspots of the EUR tracer are located north of 50°N, close to the Arctic. During summer, EA, EUR, NAM, and TPSA (Fig. 4g-j) all show the highest mixing ratio in the upper troposphere in the Arctic, consistent with the GLB tracer (Fig. 4f). As discussed in Orbe et al., (2015), these tracers are first lifted into the upper troposphere likely by convection that penetrates the isentropes, followed by horizontal transport into the Arctic. The TPSA tracer is steered into the subtropical upper troposphere-lower stratosphere by upwelling in the Asian summer monsoon region, and subsequently, transported into the Arctic lower stratosphere through the stratospheric pathway (Zheng et al., 2021), which contributes to the Arctic tracer concentration in the upper troposphere.



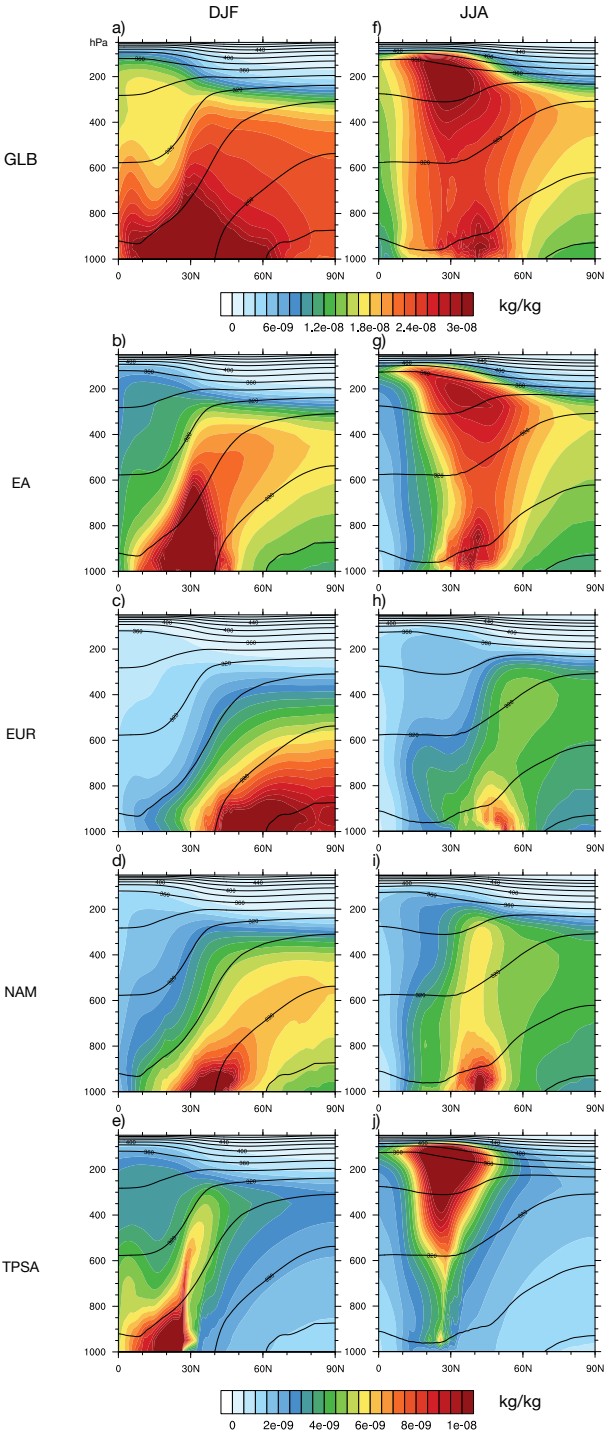

**Figure 4: Zonal mean tracer mixing ratio (kg kg$^{-1}$) during winter (a-e) and summer (f-j) for GLB, EA, EUR, NAM and TPSA tracers. The contours show the zonal mean potential temperature, with a contour interval of 20 K.**





The relative contribution from each emission region at different pressure levels averaged over the entire Arctic, in terms of percentages, is summarized in Fig. 5. During both winter and summer, the EA, EUR, and NAM tracers dominate in the troposphere, accounting for up to 80% of the total. Conversely, in the Arctic lower stratosphere (above 200 hPa), the TPSA tracer contributes the most, due to the combination of substantial surface emission over the TPSA region and the

stratospheric transport pathway as previously discussed (Orbe et al., 2015, Zheng et al., 2021). During summer (Fig. 5b), the percentage contribution across different regions remains consistent from the surface to the upper troposphere (300 hPa). However, during winter, the EUR tracer shows the largest contribution in the lower troposphere, whereas the EA and NAM tracers contribute more than the EUR tracer in the upper troposphere. This difference between summer and winter could be attributed to various transport pathways from different regions during different seasons, as discussed above (Fig. 4). In

summer, the EA, EUR, and NAM tracers share similar vertical transport pathways into the Arctic, with the horizontal transport peaking in the upper troposphere. In winter, the EUR tracer is transported into the Arctic in the lower troposphere, while the EA and NAM tracers are primarily directed into the Arctic mid-to-upper troposphere.

## Arctic C0_50 contribution (%) by different regions

a) DJF

b) JJA

East Asia (EA)

Europe (EUR)

North America (NAM)

Tibetan Plateau & South Asia (TPSA)

Middle East (ME)

Central Asia & Siberia (CAS)

Southeast Asia (SEA)

Africa (AFR)

South America (SAM)

Remaining regions (REM)





**Figure 5: Relative contribution (percentage) by different emission regions at different pressure levels averaged over the Arctic region during winter (a) and summer (b).**

## 4 Interannual Variability

We now delve into the interannual variability of tracer transport into the Arctic during winter and summer. The horizontal structure of interannual variability, quantified by the standard deviation of seasonal-mean column tracer mass at each grid

point, is depicted in Fig. 6. The three primary midlatitude source regions, the EA, EUR, and NAM tracers, contribute significantly to the interannual variability over different sectors of the Arctic. Specifically, the largest interannual variability of these three regional tracers is located over Siberia and the Pacific side of the Arctic (Fig. 6a and 6e), the Eurasia side of the Arctic (Fig. 6b and 6f), and the North Atlantic side of the Arctic (Fig. 6c and 6g), respectively. This highlights that the interannual variability over different sectors of the Arctic originates from distinct midlatitude emission source regions.

Conversely, the TPSA tracer emitted in the subtropics has a more zonally coherent interannual variability in the Arctic (Fig. 6d and 6h).

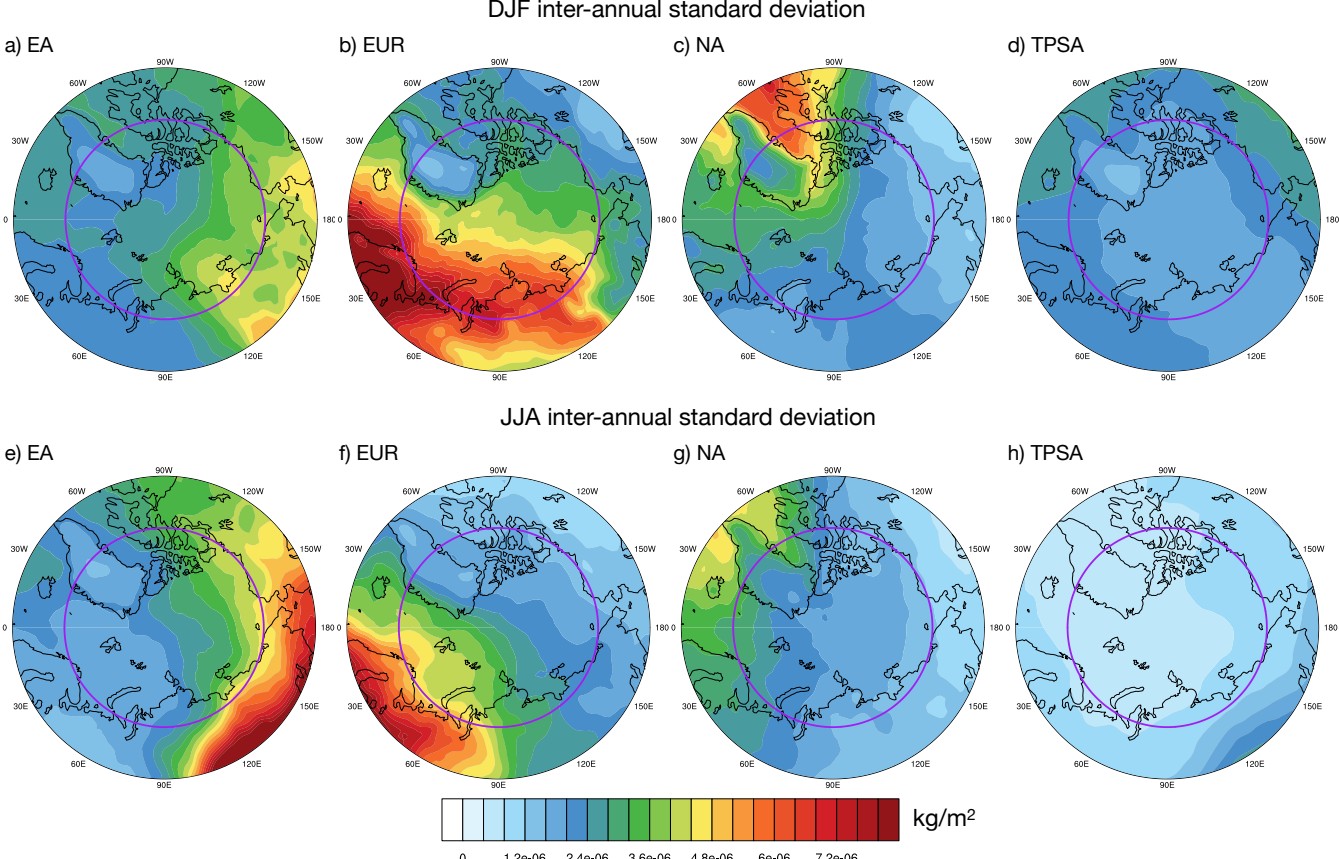

**Figure 6: Interannual standard deviation of seasonal-mean column tracer mass (kg m⁻²) at each grid point over the Arctic for EA, EUR, NAM and TPSA tracers in winter (a-d) and winter (e-h) respectively. The purple circle depicts the boundary of the Arctic**
**region (70°N).**



To further identify the interannual variability in spatial patterns due to transport into the Arctic, we apply EOF analysis on seasonal-mean column tracer mass from each regional tracer in the Arctic region (70°-90°N). The EOF analysis can provide both spatial and temporal information of the interannual variability of the tracer transport into the Arctic. For both winter and summer, we will start with tagged tracers emitted from individual regions, as interannual variability for a single emission region is easier to comprehend. After that, we will explore the interannual variability of the GLB tracer, which considers all emission regions collectively, and examine how individual emission regions contribute to the interannual variability of the GLB tracer.

**4.1 Winter Regional Tracers**

The spatial structures of EOF1 for EA, EUR, NAM, and TPSA tracers are shown in Fig. 7a-d, respectively. For all four tracers, EOF1 exhibits a monopole structure, signifying a coherent increase (or decrease) in tracer transport into the Arctic associated with EOF1. The first EOF explains a significant portion of the horizontal spatial variability, with the spatial variance explained (top-right corner of each panel; also see column 1 in Table 1) exceeding 50% for three out of the four tracers. The spatial patterns of EOF2, EOF3, and EOF4 are shown in Fig. S2, with much smaller variance explained in each case.

| DJF CO50 | EOF1 spatial (grid-point) variance explained | EOF1 (horizontally and vertically integrated) Arctic tracer mass variance explained | Correlation: EOF1 vs Arctic tracer mass |
|---|---|---|---|
| East Asia (EA) | 68.0% | 99.5% | 0.998 |
| Europe (EUR) | 39.9% | 93.6% | 0.967 |
| North America (NAM) | 51.1% | 93.2% | 0.965 |
| Tibetan Plateau and South Asia (TPSA) | 88.7% | 99.9% | 0.999 |
| Middle East (ME) | 62.3% | 98.3% | 0.992 |
| Central Asia and Siberia (CAS) | 55.1% | 96.7% | 0.983 |
| Southeast Asia (SEA) | 55.8% | 99.6% | 0.998 |
| Africa (AFR) | 62.0% | 99.9% | 0.999 |
| South America (SAM) | 52.3% | 99.1% | 0.995 |
| Remaining regions (REM) | 53.2% | 99.3% | 0.996 |

**Table 1: Summary of Arctic spatial variance explained by EOF1 for each regional tracer (column 1), Arctic tracer mass variance explained by EOF1 (column 2), and the correlation between EOF1 time series and winter seasonal-mean Arctic tracer mass time series.**







**Figure 7: (a-d) Interannual winter column tracer mass EOF1 pattern (kg m$^{-2}$) for EA, EUR, NAM and TPSA tracers, respectively.**
**The spatial variance explained by EOF1 is shown in the top-right corner. (e-h) Similar to (a-d), but for EOF1 regressed column**
**tracer mass over the northern hemisphere. The black boxes show the boundary of the emission region. (i-l) Similar to (e-h), but for**
**Z500 regression; unit in m. The purple circle depicts the boundary of the Arctic region (70°N). The arrows in (i-l) illustrate the**
**anomalous geostrophic wind regression patterns at 500 hPa. The scaling for the arrows is shown at the bottom right of the panel l),**
**unit in m s$^{-1}$.**

How important is the first EOF contributing to the total Arctic tracer mass, which is the total tracer mass integrated both horizontally and vertically within the entire Arctic explored in Sect. 3.1 and Fig. 2? The Arctic tracer mass variance associated with EOF1 can be calculated as the square of the area-weighted sum of column tracer mass in the EOF1 pattern, while the variance of Arctic tracer mass (horizontally and vertically integrated), is simply the variance of the Arctic tracer mass time series (the square of the standard deviation, which is shown in Fig. 2i-l). The ratio between the two values depicts the relative contribution of EOF1 on Arctic tracer mass, which is summarized in column 2 of Table 1. Across all emission regions, more than 93% of the variance in Arctic tracer mass is associated with EOF1, meaning that EOF1 drives almost all the variability in total Arctic tracer mass for tagged tracers emitted in individual regions. This is further supported by the high correlation between the EOF1 time series and the total Arctic tracer mass time series for each region (column 3 of Table 1). On the other hand, other EOF patterns (e.g., Fig. S2) primarily drive spatial shifts in the tracer distribution within the Arctic. When considering their contributions to Arctic tracer mass as a whole, the positive and negative values associated with the spatial shifts in the tracer distribution largely cancel, resulting in little impact on total Arctic tracer mass from these EOFs. We will focus on Arctic tracer mass (EOF1) for each individual emission region, while the spatial shift in the tracer distribution will be considered when analyzing the GLB tracer.

To understand the atmospheric transport associated with EOF1, regression maps of column tracer mass and 500 hPa geopotential height (Z500) are shown in Fig. 7e-h and Fig. 7i-l, respectively. Fig. 7a-d and Fig. 7e-h are essentially showing the same column tracer mass, but for the Arctic region and the northern hemisphere, respectively. The hemispheric maps (Fig. 7e-h) all illustrate poleward shifts of the tracer distribution in the positive phase. Specifically, EOF1 represents the shift of the tracer distribution: 1) between the extratropics and the tropics for the EA and TPSA tracers; 2) between the Mediterranean region and the Arctic for the EUR tracer; 3) between the southern part of the North Atlantic and the Arctic for the NAM tracer. The transitions from negative to positive anomalous column tracer mass all occur near the tracer emission regions in Fig. 7e-h. This means that the positive phase of EOF1 (i.e. more tracer transport into the Arctic) results from more eastward (i.e. downstream) and poleward transport over the emission region, while the negative phase of EOF1 (i.e. less transport into the Arctic) is due to more westward and equatorward transport over the emission region. This is supported by the large-scale circulation represented by Z500, as the large-scale anomalous wind associated with EOF1 (illustrated by the direction of the wind vectors) favours eastward and poleward transport over the emission region when a high amount of emitted tracer is transported into the Arctic (positive phase of EOF1). In other words, what drives the variations in Arctic tracer mass (EOF1) for individual emission regions is not the anomalous circulation patterns near the Arctic but rather those over the tracer source region, which promotes poleward transport. Note that the circulation pattern associated with EUR EOF1 (Fig. 7j) resembles the positive phase of NAO, which is consistent with findings in previous studies (Eckhardt et al.,




2003; Duncan and Bay 2004) that a positive NAO favours the transport of pollutants from Europe into the Arctic. The Z500 regression pattern for EA EOF1 (Fig. 7i) resembles the ENSO teleconnection pattern over the tropical Pacific and the eastern North Pacific, and the EA EOF1 time series is significantly correlated with the seasonal mean Niño 3.4 index (correlation coefficient of 0.43).

## 4.2 Winter GLB Tracer

How much of the interannual variability of the Arctic tracer mass from all source regions can be explained by each regional tracer? To understand the interannual variability of the GLB tracer during winter, which is the sum of all regionally emitted tracers, we apply the same EOF analysis as that for each regional tracer. The first 4 EOF patterns of the GLB tracer are shown in Fig. 8a-d. Similar to regional tracers, GLB EOF1 exhibits a monopole pattern in the Arctic, while the subsequent EOFs display spatial shifts of Arctic tracer mass distribution. The total Arctic tracer mass variance associated with each EOF is summarized at the bottom of Fig. 8e. Again, EOF1 explains almost all the Arctic tracer mass (integrated horizontally and vertically) variability of the GLB tracer, similar to the regional tracers.

To partition the contributions from each regional tracer to the GLB EOFs, we regress GLB EOF time series onto regional tracers (e.g., Fig. 9a-e). Note that the sum of the regressed regional tracer patterns (Fig. 9b-e) matches very well with the GLB EOF patterns (Fig. 9a), meaning that this regression method is an effective way to decompose the contributions from regional tracers. The contribution to total Arctic tracer mass from each regional tracer is shown in the bar plot in Fig. 8e. Almost all regional tracers make positive contributions to the GLB EOF1. For GLB EOF2, it is primarily a spatial redistribution of the GLB tracer, with minor contributions from any of the regional tracers to Arctic tracer mass. GLB EOF3 presents an opposing contribution from the ASI or EUR tracers, besides the spatial redistribution of the GLB tracer. We will explore the first 3 GLB EOFs in detail, which in total explains more than 70% of the spatial variance of the Arctic tracer mass from GLB tracers. Specifically, GLB EOF1 explains the variability of total Arctic tracer mass; GLB EOF2 is the major spatial redistribution pattern of the GLB tracers; and GLB EOF3, and to some extent EOF4, is an example of opposing contributions from different regional tracers.







**Figure 8:** (a-d) The first 4 EOF patterns of interannual winter column tracer mass pattern (kg m$^{-2}$) for GLB tracer. The spatial variance explained by EOFs is shown in the top-right corners. (e) The bar plot shows the contribution to Arctic tracer mass from each emission region in different EOFs. The spatial variance and Arctic tracer mass variance explained are summarized by the numbers below the bar plot.

### 4.2.1 Winter GLB Tracer EOF1

For EOF1, the EA tracer has the largest contribution (Fig. 8e). Despite that EUR, NAM, and TPSA tracers rank from second to fourth in the climatological contribution to Arctic tracer mass (Fig. 2a), their relative roles in contributing to GLB tracer EOF1, which determines the variability of Arctic GLB tracer mass, are reversed. The variability of Arctic GLB tracer mass



is determined by the circulation mode when different regional tracers are positively correlated (Fig. 8e). However, regional tracers (e.g., the EUR tracer) could exhibit significant negative correlation with other regional tracers (e.g., the EA tracer; see Table S1), meaning regional tracers could have cancelling effects on the variability of GLB tracer (which will be further
discussed below). Therefore, regional tracers with large climatology and interannual variability in the Arctic (e.g., the EUR tracer) do not necessarily have major contributions to the variability of total Arctic GLB tracer mass (GLB EOF1). The horizontal and vertical regression patterns of regional tracers are shown in Fig. 9. The EA and TPSA tracers, which exert the most significant influence on EOF1, demonstrate a horizontally coherent structure over the Arctic (Fig. 9b-c) and mainly impact the mid-to-upper troposphere (Fig. 9g-h). The NAM and EUR tracers only influence the Arctic regionally (Fig. 9d-e)
within the lower troposphere (Fig. 9i-j).

The regressed Z500 pattern of GLB EOF1 is shown in Fig. 9k. The positive Z500 anomaly over the tropical Indian Ocean and negative anomaly over midlatitude Asia enhance the zonal wind over the TPSA and EA region (arrows), driving more downstream transport which results in more tracer mass in the Arctic, corresponding to EOF1 of the TPSA and EA tracers shown in Fig. 7. The circulation over the North Atlantic also favours transport into the Arctic for the EUR and NAM tracers.
The correlation between GLB EOF1 time series and SST at each grid point is shown in Fig. 9m. The dipole pattern of Z500, with a positive Z500 anomaly over the tropical Indian Ocean and a negative Z500 anomaly over the Asia continent (Fig. 9k), is likely driven by the warm SST anomalies over the Indian Ocean suggested by the correlation.

Yang et al., (2019) attributed the inter-model spread of CO50 transport into the Arctic in CCMI models to the position of the Hadley cell edge, which is also represented by the meridional location of the midlatitude jet in their analysis. Further
evidence supporting this conclusion during winter has been found in an idealized dynamical model, in which the Hadley cell edge and jet locations are varied, as demonstrated in Yang et al., (2020). More specifically, in CCMI models, a more equatorward jet favours greater transport of CO50 (GLB tracer in our study) into the Arctic. The regression pattern of 500 hPa zonal wind (U500; Fig. 9l) reveals that the GLB EOF1 is also associated with an equatorward shift of the jet over Asia and the North Pacific (Fig. 9l), meaning that the mechanism driving interannual variability and inter-model spread of CO50
transport into the Arctic during winter is likely consistent. This mechanism has been further elaborated in Yang et al., (2020). The mean meridional circulation north (poleward) of the Hadley cell edge exerts strong poleward transport in the mid-to-lower troposphere (see Fig. 7 in Yang et al., 2019 and Fig. 1 in Yang et al., 2020). Therefore, the relative location of the mean meridional circulation to the emission region determines the strength of the transport into the Arctic. As pointed out by Yang et al., (2020), the midpoint of the main CO50 emission region (20°-40°N; Fig. 1) is around or south of the
Hadley cell edge during winter, meaning an equatorward shift of the mean meridional circulation associated with the southward shift of the Hadley cell edge (or the jet stream) leads to increased poleward transport of CO50 and thus higher CO50 concentration in the Arctic. Our analysis further shows that the larger amount of CO50 in the Arctic (positive GLB EOF1), driven by this equatorward shift of the jet, is indeed achieved by driving enhanced poleward transport of the EA and TPSA tracers emitted around 20°-40°N into the Arctic (Fig. 8e).




**Figure 9: (a) GLB EOF1 spatial pattern of column tracer mass. (b-e) The regression of GLB EOF1 time series onto EA, TPSA, NAM and EUR tracer column mass, respectively. (f-j) Zonal and vertical structure of GLB and regional tracers, which is shown by the regression of GLB EOF1 time series onto the meridional average (70°-90°N) of GLB, EA, TPSA, NAM and EUR tracer mixing ratio, respectively. (k) The regression of GLB EOF1 time series onto Z500. The arrows illustrate the anomalous geostrophic wind regression patterns at 500 hPa. The scaling for the arrows is shown at the bottom right of the panel, unit in m/s. (l) The regression of GLB EOF1 time series onto zonal wind at 500 hPa (U500). The darkgreen contours are winter climatology of U500, with a contour interval of 20 m s$^{-1}$. The zero contour is omitted. Positive contours are shown in the solid lines while negative contours are shown in the dash lines. m) Correlation between GLB EOF1 time series and SST and each grid point during winter. The hashed regions show where the correlation is statistically significant at 95%. The purple line represents the boundary of the Arctic in the maps.**

### 4.2.2 Winter GLB Tracer EOF2 and EOF3

EOF2 of the GLB tracer exhibits a spatial shift in tracer distribution between the Atlantic and Pacific sides of the Arctic, with the primary contribution coming from the EUR tracer in both the horizontal and vertical structures of EOF2 (Fig. 10). The associated circulation pattern (Fig. 10i) demonstrates an anomalous trough over Europe. The anomalous northerly wind west of the trough over the North Atlantic impedes transport into the Arctic, while southerly wind east of the trough over Siberia favours transport into the Arctic. These wind anomalies result in the spatial shift of the distribution of the EUR





tracer. This circulation pattern, which resembles the Arctic Oscillation (AO) associated with shifts of geopotential height between the Arctic and the midlatitudes, corresponds well to the dominant mode of the model's winter interannual variability

in the northern hemisphere (the first EOF of winter Z500 over the northern hemisphere, see Fig. S3). Note that the NAM tracer distribution (Fig. 10h) shows a vertical shift of the tracer distribution, while other regional tracers associated with GLB EOF2 and EOF3 mostly display a horizonal shift of the distribution. This vertical shift (Fig. 10h), is driven by the cyclonic circulation centred over Greenland (Fig. 10i), which suppresses the poleward transport over the Baffin Bay (west of Greenland) and enhances the cross-Atlantic transport which is more favourable in the mid-to-upper troposphere (e.g. Zheng

et al., 2021). After the cross-Atlantic transport, the tracers are further transported into the Arctic over Eurasia and covers much of the Arctic mid-to-upper troposphere, which is revealed by northern hemisphere regression maps (not shown). This results in the vertical dipole pattern of the NAM tracer associated with GLB EOF2 (Fig. 10h).

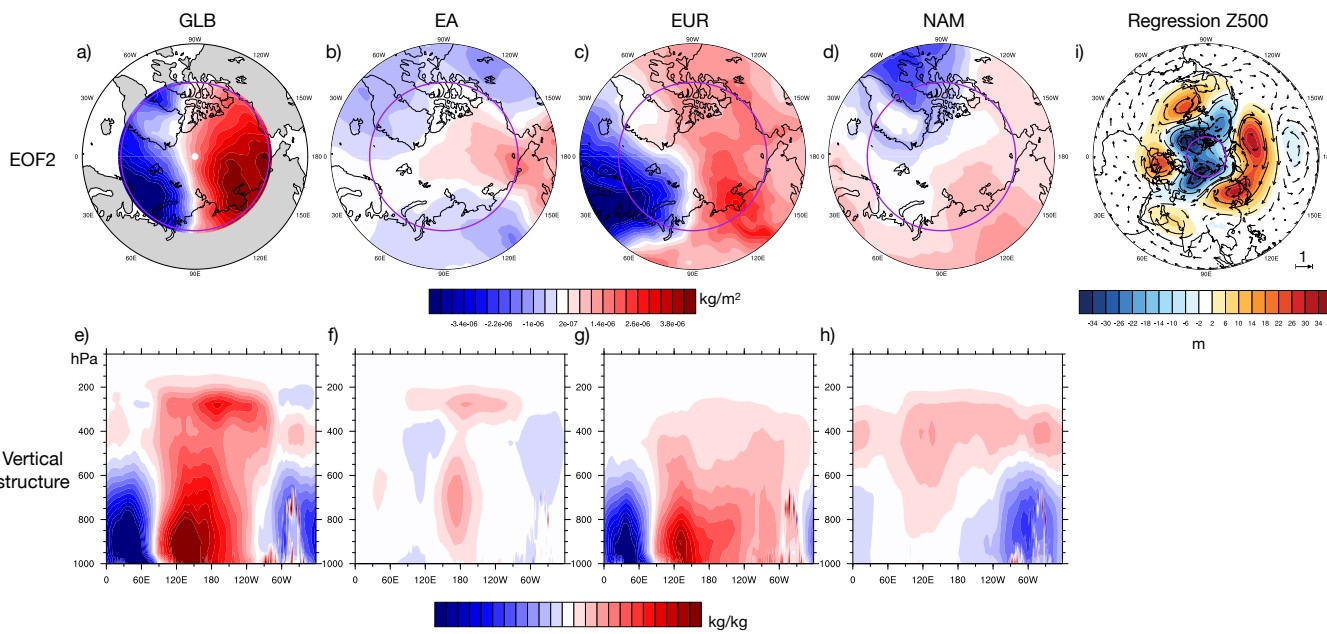

**Figure 10: (a) GLB EOF2 spatial pattern of column tracer mass. (b-d) The regression of GLB EOF1 time series onto EA, EUR and**
**NAM tracer column mass, respectively. (e-h) Zonal and vertical structure of GLB and regional tracers, which is shown by the regression of GLB EOF2 time series onto meridional average (70°-90°N) of GLB, EA, EUR and NAM tracers, respectively. (i) The regression of GLB EOF1 time series onto Z500. The arrows illustrate the anomalous geostrophic wind regression patterns at 500 hPa. The scaling for the arrows is shown at the bottom right of the panel, unit in m s⁻¹.**

EOF3 of the GLB tracer reveals a spatial shift in tracer distribution between the Eurasia and North America sides of the
Arctic, with a compensating effect on Arctic tracer mass from the EA and EUR tracers. In the positive phase of EOF3, the EA tracer contributes positively over the entire Arctic (Fig. 11b) in the mid-to-upper troposphere (Fig. 11f), while the EUR tracer has a negative contribution over the Eurasia side of the Arctic (Fig. 11c) in the lower troposphere (Fig. 11g). In addition, the contribution of the NAM tracer amplifies the EOF3 pattern. As a result, EOF3 involves both horizontal and vertical redistribution of the GLB tracer (Fig. 11e), with anomalous negative tracer concentration in the lower troposphere





over Eurasia, and anomalous positive tracer concentration in the upper troposphere and the North America side of the Arctic. The associated circulation pattern responsible for driving opposite transport of the EA and EUR tracers into the Arctic is depicted in Fig. 11j. Anomalous winds leading to the opposite transport of EA and EUR tracers result from a positive Z500 anomaly over the northern North Atlantic (centered over the Norwegian Sea), a negative Z500 anomaly over central Siberia, and a positive Z500 anomaly over midlatitude western North Pacific (centered over Japan). The Z500 pattern associated with

these three anomalies has been identified as the Eurasian teleconnection pattern (Wallace and Gutzler 1981), one of the major teleconnection patterns during winter.

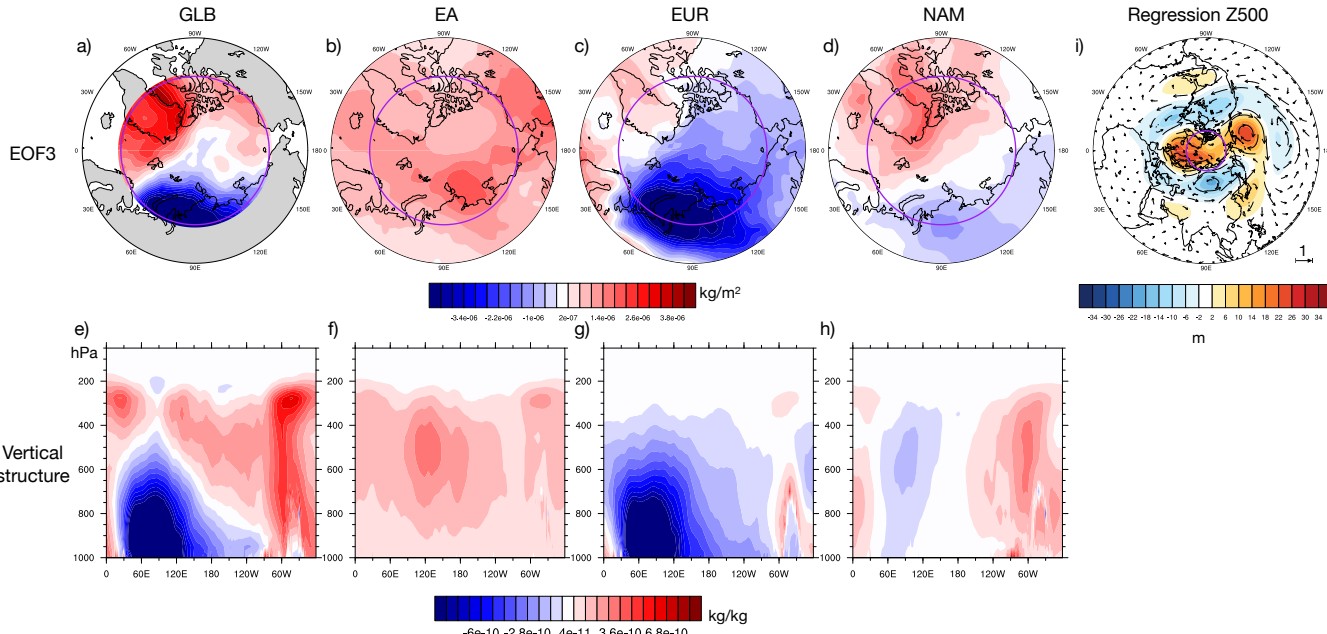

**Figure 11: Similar to Fig. 10, but for EOF3.**

**4.3 Summer Regional Tracers**

Using a similar approach to the one applied in winter, the spatial structures of summer EOF1 for the EA, EUR, NAM, and TPSA tracers, as well as the associated circulation patterns, are shown in Fig. S4. The variance explained by EOF1 in both spatial variance and Arctic tracer mass is summarized in Table S2. The major findings in summer are highly consistent with those in winter, as we see: 1) EOF1 displays a monopole pattern in the Arctic for each regional tracer (Fig. S4a-d), and

drives almost all the variability in Arctic tracer mass, as the variance of Arctic tracer mass associated with EOF1 exceeds 97%, except for the CAS tracer (Table S2); 2) EOF1 shows poleward versus equatorward shifts of the tracer distribution (Fig. S4e-h) for each regional tracer, with the transitions of the shifts consistently occurring near the tracer emission region; 3) The associated circulation patterns (Fig. S4i-l) reveal that eastward and poleward winds over the emission regions drive





the shifts of tracer spatial distribution associated with EOF1, resulting in more transport of tracer into the Arctic in the
positive phase of EOF1.

## 4.4 Summer GLB Tracer

Again, we employ the same EOF technique to understand the interannual variability of the GLB tracer during summer. The
first 4 EOF patterns of the GLB tracer are shown in Fig S6a-d. Similar to regional tracers and the winter GLB tracer, EOF1
displays a monopole pattern in the Arctic, and the subsequent EOFs show spatial shifts of tracer distribution. The Arctic
tracer mass variance associated with the EOFs is shown at the bottom of Fig. S6e. Not surprisingly, EOF1 explains more
than 80% of the variance of Arctic GLB tracer mass. However, EOF2 in summer still explains a substantial amount of the
variance (16%) of the Arctic tracer mass. Given that the first two EOFs account for more than 97% of the Arctic mass, we
will focus our analysis on these two EOFs.

Unlike GLB EOF1 and EOF3 during winter (Fig. 8), which are shaped by multiple regional tracers with similar contributions
in terms of magnitude to Arctic GLB tracer mass, EOF1 and EOF2 during summer are dominated by the EUR and EA
tracers, respectively (Fig. S6e). Therefore, the spatial patterns associated with GLB EOF1 and EOF2 (Fig. S6) in both tracer
distribution and circulation are expected to closely resemble that of regional EUR EOF1 and regional EA EOF1 (Fig. S4).
This is confirmed by the analysis similar to Fig. 9-11 in Fig. S7 and S8. The spatial pattern of GLB EOF1 tracer is primarily
contributed by the EUR tracer (Fig. S7), and the associated circulation (Fig. S7i) closely mirrors that in Fig. S4j. This
circulation pattern, which favours poleward transport of the EUR tracer, bears a strong resemblance to the NAO during
summer, which is also revealed by the second EOF of summer Z500 in this simulation (Fig. S9). Similarly, the spatial
pattern of GLB EOF2 largely comes from the EA tracer (Fig. S8), with an opposite contribution from the EUR tracer,
consistent with Fig. S6e. The circulation associated with GLB EOF2 (Fig. S8i) is also similar to that of regional EA tracer
EOF1 (Fig. S4i), enhancing eastward and poleward transport of the EA tracer near its source region.

## 445  5 Tracers with Different Lifetimes

### 5.1 Characteristics of the Transport into the Arctic

In this section, we explore how different lifetimes of tracers modulate transport into the Arctic and the relative contribution
from different emission regions. As discussed in Sect. 2, tracers with the same surface emission flux as CO50 but varying
lifetimes (100, 25, 15, 10, and 5 days) are implemented within the same simulation, mimicking the transport of chemical
species with different removal rates, and are denoted as CO100, CO25, CO15, CO10, and CO05.

The features of transport into the Arctic for these tracers in winter are summarized in Fig. 14. It is evident that the relative
contribution of the EA tracer in the Arctic decreases as the tracer lifetime becomes shorter, while the EUR tracer makes the
dominant contribution when the tracer lifetime is shorter. This aligns well with the horizontal transport features of regional
tracers discussed in Sect. 3.2.1 (see Fig. 3), as it takes a shorter distance and time for the EUR tracer to be transported into





the Arctic, while the transport pathway is the longest for the EA tracer among the top three regions with the largest contribution (EA, EUR, and NAM). Therefore, shorter lifetimes result in much of the EA tracer being removed from the atmosphere along its extended transport pathway into the Arctic, leading to the dominance of the EUR tracer in the Arctic, which has a shorter transport time scale.

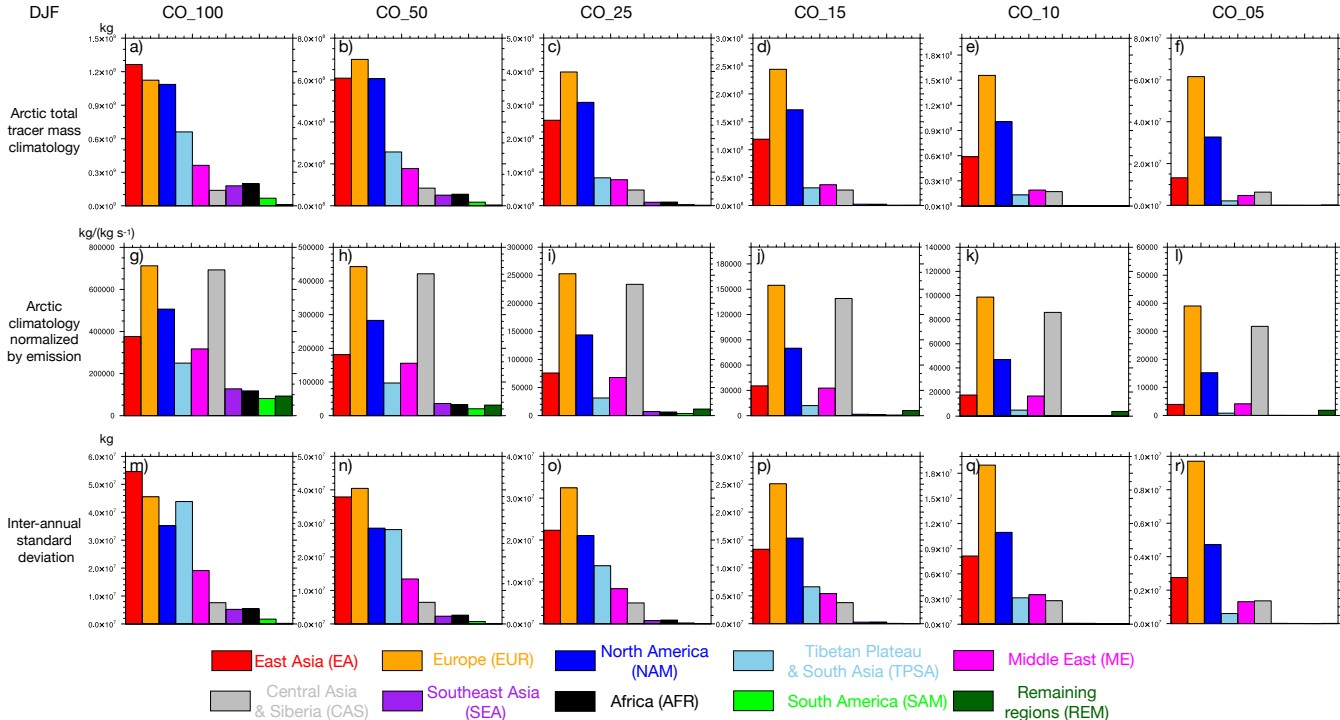

**Figure 12: (a-f) Similar to Fig. 2a, but for CO100, CO50, CO25, CO15, CO10 and CO05, respectively. (g-l) Similar to (a-f), but for Arctic tracer mass climatology normalized by emission of each region. (m-r) Similar to (a-f), but for interannual standard deviation.**

Ideally, when the tracer's lifetime is sufficiently long (for example, a few decades), the tracer becomes well-mixed in the atmosphere (the same mixing ratio everywhere), meaning the Arctic tracer mass normalized by emission (e.g., Fig. 14g-l)

becomes the same for each emission region. This tendency can be observed in Fig. 14g-l, where values from each emission region become relatively closer as the lifetime increases. In other words, with a sufficiently long lifetime, the contribution to Arctic tracer mass (e.g., Fig. 14 a-f) becomes proportional to the regional emission of the tracer. Thus, it is not surprising to find that the EA and TPSA tracers, which have large surface emissions (Fig. 1), play a more substantial role in Arctic tracer mass (Fig. 14 a-f) and the associated interannual variability (Fig. 14 m-r) when tracer lifetimes become longer. Similar

conclusions can be drawn for summer (Fig. S10).



## 5.2 Interannual Variability

Here, we briefly discuss how different lifetimes modulate the interannual variability of tracer transport into the Arctic. When considering individual regional tracers, the dominant modes driving mass contribution in the Arctic (EOF1 of regional tracers discussed in Sect 4) remain consistent across different tracer lifetimes. Taking winter as an example, consistent with the discussion in Sect 4.1, EOF1 of regional tracers explains almost all the variance of Arctic tracer mass (Table S4) regardless of tracer lifetimes. In addition, the spatial structures of EOF1 of the same regional tracer are consistent across different lifetimes (e.g., comparing CO50 and CO10 in Fig. S11), meaning the same circulation determines the mass transport into the Arctic for each regional tracer with different lifetimes. As discussed in Sect 4.1, the anomalous winds associated with EOF1 circulation over the emission region essentially drive the spatial shift in tracer distribution, resulting in different Arctic tracer mass. The wind patterns over the emission region influence the tracer transport almost immediately after the tracer is emitted regardless of the tracer's lifetime, explaining the consistency of these EOF1 patterns.

However, the interannual variability of the GLB tracer, which represents the sum of all regional tracers, exhibits varying dominant modes depending on the tracer's lifetime. As discussed above (Sect. 5.1), when the tracer lifetime is shorter, contributions from regions with shorter transport pathways (EUR) dominate. Conversely, for a longer tracer lifetime, regions with larger emissions (EA and TPSA) become the major contributors. Therefore, the dominant mode (EOF1) of the GLB tracer is mostly contributed by the EA and TPSA tracers for CO100 (not shown). In contrast, the EUR tracer dominates EOF1 of CO10 and CO05 (not shown). This highlights that the importance of different emission regions varies with different tracer lifetimes.

## 6 Conclusions

In this study, we investigate the influence of atmospheric circulation on the interannual variability of long-range transport from different emission regions into the Arctic by employing idealized tracers with spatially varying but temporally fixed emissions in WACCM6. The interannual variability (standard deviation) of Arctic tracer mass due to transport driven by atmospheric circulation is about 5-15% of the climatology. The EA, EUR, and NAM tracers make the largest contributions to Arctic tracer mass during all seasons, followed by the TPSA and ME tracers. During winter, the EUR tracer primarily influences Arctic tracer concentration in the lower troposphere; while other tracers with large contributions are first lifted into the mid-troposphere over the midlatitudes and subtropics, and then horizontally transported into the Arctic following the isentropes, making most contributions in the Arctic mid-to-upper troposphere. During summer, regionally emitted tracers with large contributions share a similar transport process, as they are first transported upward across the isentropes by convective processes over the midlatitudes and subtropics, and then into the Arctic mid-to-upper troposphere.

Interannual variability of transport into the Arctic from different regionally emitted tracers and the GLB tracers is examined by EOF analysis. For regional emitted tracers, in both summer and winter, the first EOF of each tracer not only captures the most important mode of spatial variations in the Arctic but also explains almost all the interannual variability in Arctic tracer



mass associated with that particular tracer. The spatial patterns of the first EOFs for different regional tracers exhibit significant similarity, with a poleward versus equatorward shift of the tracer distribution. The transitions of these shifts in distribution are consistently located over the regional emission regions, meaning the circulation (wind) over the emission regions drives almost all the interannual variability of Arctic tracer mass for a regionally emitted tracer. This is further confirmed by the atmospheric circulation patterns associated with these EOFs that poleward and eastward wind over the emission region favours transport into the Arctic.

Similar to regional tracers, EOF1 of the GLB tracer during winter also explains almost all of the interannual variability in Arctic tracer mass. Interestingly, the EA and TPSA tracers make the largest contributions to winter GLB EOF1. The EUR and NAM tracers, despite their large contribution in climatology, make smaller contributions. The atmospheric circulation associated with winter GLB EOF1 corresponds to an equatorward shift of the midlatitude jet when a higher amount of GLB tracer is transported into the Arctic. This is consistent with Yang et al., (2019) who found that such an equatorward or poleward shift of the jet explains the inter-model spread of transport into the Arctic among the CCMI models. The result that the EA and TPSA tracers make the largest contribution to Arctic tracer mass variability associated with the shift of the jet is also consistent with the idealized simulations in Yang et al., (2020). This shift of the jet is likely driven by the SST anomalies in the tropics and subtropics. Large scale teleconnection patterns, such as the AO and Eurasian pattern, corresponding to GLB EOF2 and EOF3, spatially redistribute tracer mass in the Arctic and modulate the transport of different regional tracers into the Arctic, as different regional tracers sometimes compensate for each other in their contribution to Arctic tracer mass from year to year. In summer, GLB tracer Arctic tracer mass is contributed by both EOF1 and EOF2, which are dominated by the EUR and EA tracers, respectively. The NAO drives the variability of the EUR tracer transport in both winter and summer, which is consistent with previous studies (Eckhardt et al., 2003; Duncan and Bay 2004).

Tracers with the same emission but different idealized lifetimes, ranging from 5 days to 100 days, are also examined. The transport into the Arctic is dominated by tracers with a short temporal transport pathway into the Arctic (e.g., the EUR tracer) when the tracer lifetime is short. When the tracer lifetime becomes longer, the role of regions with large emissions becomes more important in Arctic tracer mass. The circulation (wind) patterns that drive the variability of transport into the Arctic for individual emission regions are consistent across different tracer lifetimes, as the key wind anomalies are over the emission region, influencing the transport immediately after the tracer is emitted regardless of the lifetime.

Finally, we discuss some caveats and limitations in this study. Atmospheric teleconnection patterns appear to play a crucial role in driving the variability of tracer transport, particularly for the GLB tracer. Specifically, the enhanced transport of EA and TPSA tracers into the Arctic associated with the positive phase of GLB EOF1 is likely driven by atmospheric teleconnections forced by tropical SST; spatial redistribution of the tracers over the Arctic associated with GLB EOF2 and EOF3 is driven by AO and the Eurasia teleconnection pattern. However, our 49-year WACCM6 run is not a very long simulation, meaning certain circulation variability or teleconnection patterns that may modulate the transport might not be well represented. Longer simulations would be beneficial in future studies. The detailed structures of teleconnection patterns

influencing the transport into the Arctic may be model dependent and could differ from observations. This could be further analyzed in future studies.

The emission employed in the model simulation, which is the annual mean anthropogenic emission with no temporal
variations, does not have a seasonal cycle which may influence the results. Further, the anthropogenic emission employed in the model simulation may not capture the full picture of interannual variability. Taking CO as an example, the interannual variability of CO over the Arctic during summer is largely driven by the variability in biomass burning emissions (Monks et al., 2012), particularly emissions associated with forest fires over regions close to the Arctic, including Alaska, Canada, and Siberia. Our current analysis does not highlight the transport characteristics of these regions, especially for Alaska. Future
studies employing tagged emissions associated with biomass burning would improve our understanding of how atmospheric circulation influences the transport of biomass burning generated tracers into the Arctic.

**Data availability**

Seasonal mean data of different tracers, as well as geopotential height and wind data at 500 hPa will be uploaded to Columbia University Academic Commons (free access) once the manuscript is accepted.

**Supplement**

The supplement related to this article is available online at:

**Author contributions**

CZ, YW and CO designed the model simulation. CZ performed the model simulations, analyzed the results, and wrote the paper. YW and MT supervised the project. YW, MT and CO participated in paper editing.

**Competing interests**

The authors declare that they have no conflict of interest.

**Acknowledgements**

We acknowledge the support from the National Science Foundation (NSF) Award OPP-1825858. Computing and data storage resources, including the Cheyenne supercomputer, were provided by NCAR's Computational and Information
Systems Laboratory, which is sponsored by NSF. The SST and SIC boundary forcing data, along with surface emission data, were also provided by NCAR.



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
