# Peer review of "Influence of Atmospheric Circulation on the Interannual Variability of Transport from Global and Regional Emissions into the Arctic"

_EGUsphere, 2024_

## Referee Comment (RC2)

**Review of "Influence of Atmospheric Circulation on the Interannual Variability of Transport from Global and Regional Emissions into the Arctic" by Cheng Zheng et al. (egusphere-2024-253)**

The study by Cheng Zheng and co-authors presents an analysis on transport pathways of surface emitted air pollutants into the Arctic. They employ a chemistry circulation model to assess the contribution of various source regions (with temporally fixed emissions) to the Arctic tracer mass of an idealized carbon monoxide like tracer. They come to their conclusion through analysis of tracer mass distribution in the Arctic, defined as the latitudes between 70-90 °N. More so they use empirical orthogonal functions to analyze the source of variability of the tracer mass in the Arctic. They find that the distance between the source region to the Arctic plays a crucial role for the contribution of the individual sources for the Arctic tracer mass. Another finding is that the circulation at the source region strongly impacts the contribution of the individual source regions. Sensitivities of the tracer lifetime also revealed that shorter pathways enhance the contribution of short lived species while regions with higher emissions can contribute on longer time scales to the Arctic tracer mass.

The Arctic is without doubt one of the regions most affected by current climate change. While this is obvious through temperature records, there is also strong evidence that trace species can have direct and indirect impacts on the Arctic climate. Therefore it is most important to understand how anthropogenic emissions contribute to the Arctic composition. This study directly addresses this question through a thorough analysis. The questions of the study are clearly stated. Setup of the model experiments are clearly communicated, figures are very good and the answers to the questions are based on the analysis and well justified. The paper is well written and the line of thought comes out clearly to me. The paper is in my opinion well suited for publication in ACP and I have in general only minor comments. These are outlined below.

**Comments**

- Definition of the "Arctic" (p.5, l.128): Is there a reason for 70°N-90°N as the Arctic? The authors also mention the polar dome? Does this latitude band reflect this dome like structure?

- Column tracer mass (p5, l.130): Is there a reason that it is the mass up to 10 hPa? Is the highest model level where tracer mass is found?

- Abbreviations (all sections): In the manuscript many abbreviations are used and I am bad at remembering these. Maybe an overview table would help here some readers (just a suggestion).

- "downstream from the jet stream" (p7, l 169): I could not really make sense out this term. What do you mean here? Is it downstream of  a jet stream maximum or

downstream along the jet stream? Clearly, the jet transports tracer downstream, but also the jet can be found (almost) at every longitude in the extratropics.

- Section 3.2.2 Vertical structure: I like that an analysis of the vertical structure is part of the analysis and that the transport pathways come out as expected. I wondered if it would be possible to extend the analysis slightly. For instance, I wonder if the pathways could be distinguished more quantitatively. Is there substantial transport near the surface during winter? How much is transported within the boundary layer, free troposphere, upper troposphere and via a stratospheric pathway? In principle most of questions could be answered by some more detailed discussion of Figure 5 which I would highly appreciate.

- Methods of Section 4: The entire Section 4 is based on the EOF analysis. However, in Section 2.3 this is only introduced very briefly. I think the authors could provide a little more information on their EOF analysis in Section 2.3, this would allow readers to more easily understand the results.  It could even stated which data analysis package has been used (if one has been used).

- Figures: The figures are overall of good quality. One tiny comment: they would be even better if the tick labels would be increased.

---

## Author Comment (AC1)

Comments on the manuscript" Influence of Atmospheric Circulation on the Interannual Variability of Transport from Global and Regional Emissions into the Arctic" by Zheng et al.

The manuscript is well written. It reports the influence of global and regional emissions on the Arctic. It is suitable for the Journal and can be published. I suggest minor changes before publication.

Thanks for your comments!

In the introduction section, the references used are old. Recent references should be mentioned. There are several papers on trajectory analysis of atmospheric transport to the Arctic. The results from these studies are also interesting.

We will add a few sentences (see below) in the introduction to discuss the utility of back trajectory analysis in understanding the transport into the Arctic.

"The observed concentration of trace gases and aerosols is influenced by emission, transport and removal processes. One way to disentangle their respective roles, specifically to isolate the role of long-range atmospheric transport, is the back-trajectory analysis which is often carried out with reanalysis or model meteorological fields to identify source regions of trace gases or aerosols in the Arctic. Such back-trajectory analysis has been applied to understand how variability in emissions and transport modulates Arctic aerosol and trace gases (e.g., Huang et al., 2010; Hirdman et al., 2010; Schmeisser et al., 2018; Leaitch et al., 2018), chemistry processes along the transport into the Arctic (e.g., Matsui et al., 2011; Gilardoni et al., 2023), as well as comparing with aircraft measurements (e.g., Willis et al., 2019; Schulz et al., 2019)."

The results presented in the study are based on WACCM6 model simulations. There may be biases in model transport processes that vary with the model.  It will affect the atmospheric circulation. You may mention it.

We will revise the sentence regarding the caveats of model simulations as "The detailed structures of the teleconnection patterns influencing the transport into the Arctic may be model dependent and could differ from observations due to model biases."  This will emphasize the possible biases in the model.

A schematic depicting overall results is needed since the result section is descriptive and quantitative.

Thanks for the suggestion! A schematic is quite helpful to visualize our results. The schematic figure below summarizing how circulation anomalies modulate the transport from different emission regions will be added to the revised manuscript.

[Figure]

Figure R1: Schematic figure summarizing the atmospheric circulation anomalies that modulate the tracer transport into the Arctic from major emission regions. The shading in the background layer shows surface emission (kg/m$^2$/s) similar to Figure 1 (values smaller than 1e-12 are not shown). The solid black lines depict the boundary of the major emission regions, including EA, TPSA, EUR and NAM. The dashed black lines represent the boundary of the Arctic region (70°N). The magenta and blue arrows show the wind anomalies favouring and unfavouring tracer transport into the Arctic for different emission regions, respectively. The magenta and orange shadings show the anomalous tracer transport (positive anomaly of column tracer mass) associated with circulation favouring tracer transport into the Arctic. Similarly, the blue and cyan shadings represent the anomalous tracer transport associated with circulation unfavouring tracer transport into the Arctic. The transport anomalies are summarized from the results for winter in Figure 7.

Conclusion section is lengthy. Concise bullets points of the important results should be given.

Following the reviewer's suggestion, we summarized the findings from the EOF analysis into 4 major points (see below), which will be included in the revised manuscript.

• For regional emitted tracers in both summer and winter as well as the global emitted tracer in winter, the first EOF of each tracer not only captures the most important mode of spatial variations in the Arctic but also explains almost all the interannual variability in Arctic tracer mass associated with that particular tracer.

- The spatial patterns of the first EOFs for different regional tracers exhibit significant similarity, with a poleward versus equatorward shift of the tracer distribution. The transitions of these shifts in distribution are consistently located over the regional emission regions, meaning that the circulation (horizontal wind) over the emission regions drives almost all the interannual variability of Arctic tracer mass for a regionally emitted tracer. This is further confirmed by the associated atmospheric circulation patterns as poleward and eastward wind over the emission region favours transport into the Arctic. Fig. 13 shows a schematic diagram summarizing the circulation anomalies and the associated tracer column mass anomalies that favour or unfavour the transport into the Arctic.

- The EA and TPSA tracers make the largest contributions to winter GLB EOF1. The EUR and NAM tracers, despite their large contribution in climatology, make smaller contributions. The atmospheric circulation associated with winter GLB EOF1 corresponds to an equatorward shift of the midlatitude jet when a higher amount of GLB tracer is transported into the Arctic. This is consistent with the findings in Yang et al., (2019) and Yang et al., (2020). This shift of the jet is likely driven by the SST anomalies in the tropics and subtropics.

- Large scale teleconnection patterns, such as the AO and the Eurasian pattern, corresponding to GLB EOF2 and EOF3 in winter, spatially redistribute tracer mass in the Arctic and modulate the transport of different regional tracers into the Arctic, as different regional tracers sometimes compensate for each other in their contribution to Arctic tracer mass from year to year. The NAO drives the variability of the EUR tracer transport in both winter and summer, which is consistent with previous studies.

---

## Author Comment (AC2)

The study by Cheng Zheng and co-authors presents an analysis on transport pathways of surface emitted air pollutants into the Arctic. They employ a chemistry circulation model to assess the contribution of various source regions (with temporally fixed emissions) to the Arctic tracer mass of an idealized carbon monoxide like tracer. They come to their conclusion through analysis of tracer mass distribution in the Arctic, defined as the latitudes between 70-90 °N. More so they use empirical orthogonal functions to analyze the source of variability of the tracer mass in the Arctic. They find that the distance between the source region to the Arctic plays a crucial role for the contribution of the individual sources for the Arctic tracer mass. Another finding is that the circulation at the source region strongly impacts the contribution of the individual source regions. Sensitivities of the tracer lifetime also revealed that shorter pathways enhance the contribution of short lived species while regions with higher emissions can contribute on longer time scales to the Arctic tracer mass.

The Arctic is without doubt one of the regions most affected by current climate change. While this is obvious through temperature records, there is also strong evidence that trace species can have direct and indirect impacts on the Arctic climate. Therefore it is most important to understand how anthropogenic emissions contribute to the Arctic composition. This study directly addresses this question through a thorough analysis. The questions of the study are clearly stated. Setup of the model experiments are clearly communicated, figures are very good and the answers to the questions are based on the analysis and well justified. The paper is well written and the line of thought comes out clearly to me. The paper is in my opinion well suited for publication in ACP and I have in general only minor comments. These are outlined below.

Thanks for your comments!

**Comments**

Definition of the "Arctic" (p.5, l.128): Is there a reason for 70°N-90°N as the Arctic? The authors also mention the polar dome? Does this latitude band reflect this dome like structure?

The reason to select 70°N as the southern boundary of the Arctic is to exclude any emission hotspots within the defined Arctic region. Note that there are a few hotspots north of 60°N in Europe. We will add a few words in the manuscript to point this out.

Prior to using 70°N as the southern boundary of the Arctic, we also tested using 60°N as the southern boundary of the Arctic. Most of the results are very similar to that using 70°N as the boundary, except for the EUR tracer, which exhibits large interannual variability between 60°N and 70°N over the Eurasian continent. It seems to be more reasonable to categorize this region (60°-70°N over the Eurasian continent) as part of the Eurasia rather than part of the Arctic, which is not a region of interest in this study as we focus on long-range transport into the Arctic. Thus, 70°N is selected as the southern boundary of the Arctic.

The polar dome structure is largely captured as the polar front and the surface/lower troposphere boundary of the polar dome is more or less around 70°N (e.g. Bozem et al., 2019). We will revise the discussion regarding Fig. 4 to include more details regarding the polar dome:

"This is reasonable given many of the emission hotspots of the EUR tracer are located north of 50°N, and some are even located north of 60°N. As the emission is close to the Arctic front, which is often depicted by strong horizontal temperature gradients and considered as the boundary of the "polar dome" in the lower troposphere, the EUR tracer is more easily transported into the "polar dome" when synoptic-scale weather systems disturb the Arctic front which acts as a transport barrier (e.g., Bozem et al., 2019)."

Column tracer mass (p5, l.130): Is there a reason that it is the mass up to 10 hPa? Is the highest model level where tracer mass is found?

Since the atmosphere between the surface and 10 hPa covers 99% of the mass of the entire atmosphere, the airmass above 10 hPa has little influence on the column integral of tracer mass. Further, from the vertical distribution of the tracers (Fig. 4), the tracer concentration is very low above 50 hPa (compared with that in the troposphere). Thus, the contribution of the tracer mass above 10 hPa is way smaller than 1% of the total column mass from the surface to model top, which would not affect our conclusions reached by using total column tracer mass.

We will add a few words in the revised manuscript to point out that the integral from the surface to 10 hPa covers 99% of the mass of the atmosphere.

Practically, performing the integral from the surface to 10 hPa only requires 35 vertical levels in WACCM6, which is only about half of the total number of model vertical levels (which is 66). Since the entire output of our 49-year simulation is more than 100TB, it is more convenient and computationally efficient to only integrate up to 10 hPa (which also has very little impact in the results). Also, we are not interested in tracers above 10hPa (in the upper stratosphere and above).

Abbreviations (all sections): In the manuscript many abbreviations are used and I am bad at remembering these. Maybe an overview table would help here some readers (just a suggestion).

We understand that the abbreviations referring to different regions could be a bit confusing. To help the readers, we will make changes in labels and titles of the figures so that the full names of the regions as well as the abbreviations are shown in the labels/titles in Figures 2-12. This will help remind readers which regions the abbreviations are referring to. It also seems to be a bit better than adding a table, as it is easier to find the information in an adjacent figure rather than look for a table at the beginning of the paper while reading.

"downstream from the jet stream" (p7, l 169): I could not really make sense out this term. What do you mean here? Is it downstream of a jet stream maximum or downstream along the jet stream? Clearly, the jet transports tracer downstream, but also the jet can be found (almost) at every longitude in the extratropics.

We will replace "by eastward transport downstream from the jet stream in the extratropics" by "by eastward transport downstream _**following**_ the jet stream in the extratropics". Here, we mean transport along the jet stream.

Section 3.2.2 Vertical structure: I like that an analysis of the vertical structure is part of the analysis and that the transport pathways come out as expected. I wondered if it would be possible to extend the analysis slightly. For instance, I wonder if the pathways could be distinguished more quantitatively. Is there substantial transport near the surface during winter? How much is transported within the boundary layer, free troposphere, upper troposphere and via a stratospheric pathway? In principle most of questions could be answered by some more detailed discussion of Figure 5 which I would highly appreciate.

Thanks for the comment! Following the suggestion, we add subpanels in Figure 5 (which are also shown below) showing the vertical profile of the GLB tracer in the Arctic, which better represents the transport into the Arctic at different vertical levels in a quantitative way.

We will revise the discussion regarding Figure 5 in the manuscript as:

"The relative contribution from each emission region at different pressure levels averaged over the entire Arctic, in terms of percentages, is summarized in Fig. 5. The vertical profiles of the GLB tracer mixing ratio, which represent the total contribution of all regional tracers, are shown in the subpanels on the right. During both winter and summer, the EA, EUR, and NAM tracers dominate in the troposphere, accounting for up to 80% of the total. Conversely, in the Arctic lower stratosphere (above 200 hPa), the TPSA tracer contributes the most, due to the combination of substantial surface emission over the TPSA region and the stratospheric transport pathway as previously discussed (Orbe et al., 2015, Zheng et al., 2021). The role of the stratospheric pathway is substantially stronger during summer, but its contribution in terms of tracer mass transport is still small compared to that in the troposphere. During winter (Fig. 5a), the transport of the GLB tracer is maximized near the surface but does not exhibit large variations from the surface to the middle troposphere. However, the relative contribution from different regional tracers varies substantially across different vertical levels. The EUR tracer shows the largest contribution in the boundary layer and lower troposphere (accounting for about 40% of the total), whereas the EA and NAM tracers contribute more than the EUR tracer in the upper troposphere. During summer (Fig. 5b), despite that the transport of the GLB in the upper troposphere is much stronger than that in the lower troposphere, the percentage contribution across different regions remains consistent from the surface to the upper troposphere (300 hPa). This means different tracers share a similar structure of vertical profiles in the troposphere, and the peak at the upper troposphere is not contributed by one particular regional tracer. The difference between summer and winter could be attributed to various transport pathways from different regions during different seasons, as discussed above (Fig. 4). In summer, the EA, EUR, and NAM tracers share similar vertical transport pathways into the Arctic, with the horizontal transport peaking in the upper troposphere. In winter, the EUR tracer is transported into the Arctic in the lower troposphere, while the EA and NAM tracers are primarily directed into the Arctic mid-to-upper troposphere."

[Figure]

Figure A1: Relative contribution (percentage) by different emission regions at different pressure levels averaged over the Arctic region during winter (a) and summer (b). The subpanels on the right of each panel show the vertical profile of Arctic averaged tracer mixing ratio (kg/kg) of the GLB tracer.

Methods of Section 4: The entire Section 4 is based on the EOF analysis. However, in Section 2.3 this is only introduced very briefly. I think the authors could provide a little more information on their EOF analysis in Section 2.3, this would allow readers to more easily understand the results. It could even stated which data analysis package has been used (if one has been used).

We will add the following sentences in the revised manuscript to provide more information about the EOF analysis:

"The EOFs are derived by computing the eigenvectors and eigenvalues of a spatially weighted anomaly covariance matrix of a field. The leading eigenvectors capture the dominant spatial patterns of variability in this field, and the corresponding eigenvalues provide a measure of percent variance explained by each mode (spatial pattern). Most of the variability in atmospheric fields is usually captured by a few leading modes. The EOF analysis is conducted on the seasonal mean time series of column tracer mass over grid boxes within the Arctic region (north of 70°N), by using the NCAR Command Language (NCL)."

Figures: The figures are overall of good quality. One tiny comment: they would be even better if the tick labels would be increased.

Thanks for the comment! We will make the tick labels in Figures 3-11 larger in the manuscript.